# Towards Trustworthy MetaShopping: Studying Manipulative Audiovisual Designs in Virtual-Physical Commercial Platforms

Esmée Henrieke Anne de Haas
The Hong Kong Polytechnic University
Hong Kong SAR
ehadehaas@connect.polyu.hk

Lik-Hang Lee*
The Hong Kong Polytechnic University
Hong Kong SAR
lik-hang.lee@polyu.edu.hk

Yiming Huang
The Chinese University of Hong Kong
Hong Kong SAR
yhuangdl@link.cuhk.edu.hk

Carlos Bermejo
Hong Kong University of Science and Technology
Hong Kong SAR
csbermejo@ust.hk

Pan Hui
Hong Kong University of Science and Technology (Guangzhou)
China
panhui@ust.hk

Zijun Lin
London School of Economics and Political Science
United Kingdom
z.lin37@lse.ac.uk

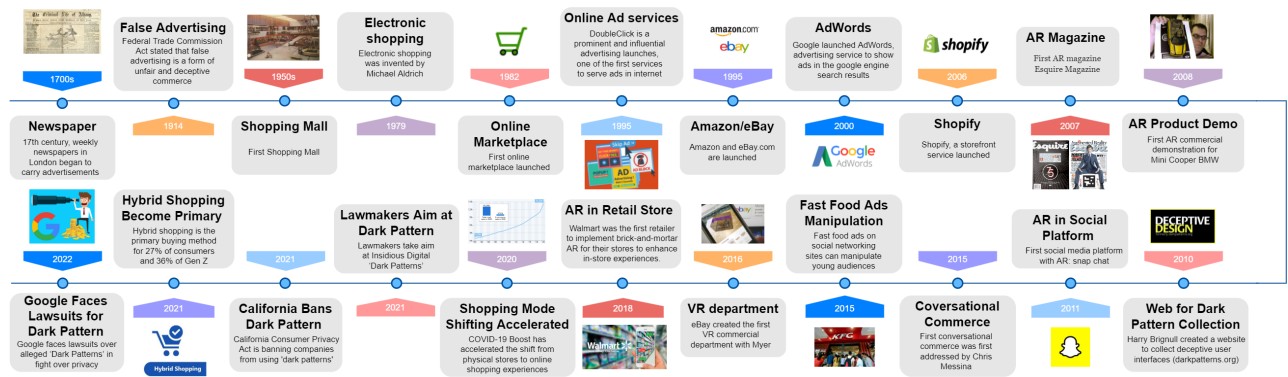

Figure 1: Timeline of e-commerce and significant regulatory responses to deceptive/manipulative designs.

## ABSTRACT

E-commerce has emerged as a significant endeavour in which technological advancements influence the shopping experience. Simultaneously, the metaverse is the next breakthrough to transform multimedia engagement. However, under such situations, deceptive designs aimed at deceiving users into making desired choices might be more successful. This paper proposes the design space of manipulative techniques in e-commerce applications for the metaverse. We construct our arguments by evaluating user interaction with manipulative design in metaverse shopping experiences, followed by a survey among users to understand the effect of counteracting manipulative e-commerce scenarios. Our findings can reinforce understanding of design guidelines according to metaverse e-commerce experiences and the possibility of opportunities to improve user awareness of manipulative experiences.

## CCS CONCEPTS

• **Human-centered computing → Mixed / augmented reality**; **Virtual reality**; **HCI theory, concepts and models**.

## KEYWORDS

Dark patterns, Empirical studies, Augmented reality, E-commerce.

**ACM Reference Format:**
Esmée Henrieke Anne de Haas, Lik-Hang Lee, Yiming Huang, Carlos Bermejo, Pan Hui, and Zijun Lin. 2024. Towards Trustworthy MetaShopping: Studying Manipulative Audiovisual Designs in Virtual-Physical Commercial Platforms. In *Proceedings of the 32nd ACM International Conference on Multimedia (MM '24), October 28-November 1, 2024, Melbourne, VIC, Australia.* ACM, New York, NY, USA, 10 pages. https://doi.org/10.1145/3664647.3681679

---

*(1) L.-H. Lee is the corresponding author and the project's PI. (2) E. H. A. de Haas and Y. Huang were interns/students at L-.H. Lee's Lab during the study.

---

## 1 INTRODUCTION

Neal Stephenson first introduced the idea of the metaverse [22], a paradigm where people, either in the physical or digital world, can interact. The metaverse is known as a digital realm [12] enabled by immersive technologies, including Virtual Reality (VR),

and Augmented Reality (AR). Recent technological advancements, such as machine learning, enable individuals to create personalised avatars [15] that can inhabit the metaverse and have their preferred voice [24]. Simeone *et al.* [51] show that real-world scenarios, such as babysitting, lecturing, and pet monitoring (e.g., dogs) can be engaged in the form of virtual interaction. E-commerce has grown steadily in recent years, and its adoption is tightly linked with technological advances. E-commerce presents several advantages to traditional retail stores. Customers of e-commerce can compare products more efficiently and make better decisions that match their desires and needs [41].

With the development of immersive technologies, people find that these technologies can help transfer e-commerce to a real-world equivalence [46]. Referring to Figure 1, people embedded AR advertisements into magazines [16] for the first time in 2007. We can also see the use of AR applications to showcase, for example, furniture (Ikea Place) and clothes (Wanna Kicks). Since *Facebook* (after rebranding as *Meta*) announced its metaverse plan, many commercial brands, e.g., Nike [29], claim to enter the metaverse era [32] by strengthening their virtual-physical media as sales channels. Similar to the impact of the Web on customers' shopping experiences, the metaverse will be a game changer in e-commerce ecosystems.

However, vendors could exploit the increased immersivity and interactivity of the metaverse to create more sophisticated manipulative designs (MDs) aiming to derail users and make suboptimal purchase decisions. The levels of misleading and surreptitiousness in MDs can be exaggerated by AR [42]. These manipulative designs, also known as dark patterns (DPs) [40], have been broadly reported in the research community [7, 17, 18, 40], appeared in online advertisements, mobile applications, and e-commerce [11, 18, 39]. Despite the challenges in perceiving and regulating these manipulative/deceptive designs, we observe that regulators, such as the California Consumer Privacy Act (CCPA), are actively protecting users against such malpractices. We can foresee that these manipulative designs in the metaverse will bring more challenges to protecting the users against them.

The paper contributes to exploring manipulative designs in virtual environments of e-commerce in the metaverse era. Our initial study (Section 3) reveals that AR commercial applications (apps) leverage changes in colours/textures of augmented objects and brightness/lighting of background environments to induce unintentional or unnecessary purchases by consumers. Accordingly, we analysed MDs in-depth that take full advantage of the immersive metaverse (e.g., realism and multi-modal cues), e.g., visuals and audio, in an augmented e-commerce scenario. It is worthwhile to mention that audio feedback has been widely used in advertisements and retail stores to influence consumers' behaviour [43]. After we describe our study design, forty participants with two AR apps in our user study demonstrate that audio feedback as MDs can significantly influence users' purchasing decisions (Section 4). Our results show that 67.5% of the participants are affected by the manipulative designs (audio-based), even when they (59%) can notice the manipulative technique. These results highlight the importance of addressing the challenges of MDs in e-commerce applications in the metaverse, followed by countermeasures with a further user survey and scenario construction (Section 5) and several design guidelines (Section 6).

## 2 RELATED WORK

**Manipulative Designs** While the metaverse will bring customers a new shopping experience, it also opens new possibilities for manipulative designs (MDs). These manipulative designs (also known as dark patterns) have been studied in several works [7, 9, 17, 17, 40]. Following Gray *et al.* [17]'s taxonomy, we can enumerate some examples of manipulative designs: *nagging,* repeated intrusions, and disturbances that redirect the users' interactions; *obstruction,* obstacles that difficult the completion of a user's action; *sneaking,* hiding or disguising reveal of information (e.g., costs); *interface interference,* interface designs that prioritise specific actions (e.g., highlighting buttons); and *forced actions,* it requires the user to do an action to access certain functionalities (e.g., registering before accessing the content). These examples are widely used in e-commerce applications and websites [11, 39, 42], where online retailers manipulate the interface and shopping procedure to influence customers in their purchase decisions. Thus, we can foresee how the current threats will become more challenging to perceive and regulate.

**Virtual entities - virtual agents manipulation.** The link between emotions and individuals can be stronger due to the immersivity of AR/VR and the possibility of pervasive interactions. The market for smart speakers and other devices that support voice assistants is growing [14]. The devices using voice assistants could use text-mining and sentiment analysis techniques to gather data that will be used to personalise the ads on websites [28, 37]. Another example is the virtual shopping assistant that helps customers have a specific tone that they like [26]. Manipulation in amplitude and frequency could influence how people like this person [55].

**Virtual entities - environment manipulation.** Augmented views, e.g., AR and augmented virtuality (AV), allow shoppers to create a more dynamic environment surrounding the customers. We could imagine a physical store that displays discounts and descriptions of targeted products in the user's field of view of their smart glasses. Another possibility is the use of location-based triggered interactions (proxemic interactions [18]) when the users move around virtual or physical retail. These manipulations of the environment will influence the users' purchase decisions [18, 42].

**User feedback manipulation.** Senses will play an essential role in the metaverse and e-commerce experience [10], where products can be not only seen but touched [1, 21, 50], tasted [25, 33, 48] and smelled [20, 47]. In particular, users view haptics through their cutaneous and kinesthetic systems, enabling the point of view regarding the material properties of surfaces and objects while considering the users' movement of their bodies. These systems are connected to their receptors, e.g., on the body and the muscles, and the tendons for the kinesthetic system [44]. These haptics could also be used to manipulate the users' movements [35], and emotional states of customers [5] in virtual worlds.

## 3 MANIPULATION OF EXISTING APPS

This section aims to probe the current manipulative designs (if any) that brands and companies implement in their mobile augmented reality (MAR) apps. Subsequently, we depict the data collection procedure and our findings regarding commercially available MAR apps for Android and iOS devices.

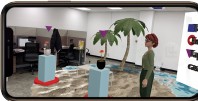 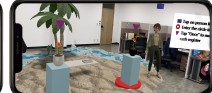 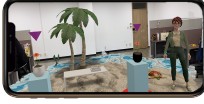 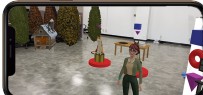 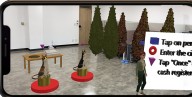 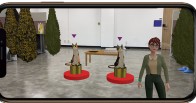

(a) App1, flowerpots.  (b) App1, flowerpots.  (c) App1, flowerpots.    (d) App2, cats.    (e) App2, cats.    (f) App2, cats.

Figure 2: MAR apps for our user study: (a–c) App 1: flowerpots; and (d–f) cats (Demo video in [38]).

**Table 1: MAR app surveyed that contains some MDs noticeable by the reviewers, where * denotes in-app brand use.**

| Name | Category | OS | Description |
|------|----------|-----|------------|
| Ideofit | Clothing/accessories | iOS/Android | Glasses |
| IWC | Clothing/accessories | iOS/Android | Watches |
| Wanna Kicks | Clothing/accessories | iOS/Android | Sneakers |
| Sephora | Beauty | iOS/Android | Lipstick/ makeup |
| Coolblue | Furniture/interior | iOS/Android | TV's/electronics |
| Ikea place | Furniture/interior | iOS | Furniture |
| Snapchat (SC) * | Clothing/accessories | iOS/Android | Farfetch lens in SC |
| Snapchat * | Clothing/accessories | iOS/Android | Target lens in SC |
| Snapchat * | Clothing/accessories | iOS/Android | Crocs lens in SC |
| Snapchat * | Clothing/accessories | iOS/Android | Gucci lens in SC |
| Snapchat * | Clothing/accessories | iOS/Android | Valentino lens in SC |
| Snapchat * | Clothing/accessories | iOS/Android | Poshmark lens in SC |
| Wonderwall AR | Furniture/interior | iOS/Android | Wallpaper |

**Data collection – App Examinations.** We focus on apps with the following features: (i) AR experience, (ii) shopping features (users can buy the displayed items), and (iii) trending online websites due to their AR features. Due to their popularity, we choose free-to-use apps and the most trending apps (top-10) under three major categories: (1) *furniture/interior*, most apps contained furniture, decoration, or electronics that could be virtually placed in a room. (2) *clothing/accessories* offered options to try out clothing or other accessories on a person virtually. (3) *Beauty* refers to different types of makeup that can be tried on a person's face. Three researchers examined the aforementioned types of MAR apps, resulting in 42 iOS and Google Play apps[1]. Some apps are not available due to geolocation restrictions (e.g., the availability of a given app may vary across different regions' app stores), which limited the selection process. Moreover, several apps are only available on either Android or iOS. As shown in Table 1, we spotted a total of 13 records of manipulative designs in the AR interfaces of the apps.

**Methodology.** Previous studies [11, 17, 39, 40, 56] explored taxonomies of manipulative design (MD) patterns and analysed their approaches. Designers used manipulative design patterns in their websites [39] and mobile apps [11]. However, very minimal efforts have been specified on manipulative or misleading designs in MAR apps. Thus, we study different brands and e-retailers that use AR in their apps to promote their products and evaluate whether we can find any manipulative design patterns or misleading interfaces during the AR experience. The three researchers searched for any odd-looking feature(s) (e.g., manipulative design) in the selected MAR apps. Each app was used for a duration of 5–7 minutes, where

the researchers performed the following tasks according to their intended use: (i) open the app, (ii) create an account or login (if necessary), (iii) follow any tutorial (if any), (iii) select two to three items to display in the AR interface, (iv) for each item move the camera and perspective of the virtual object, (v) each of the objects is overlaid under the same surface (e.g., face, floor, wall, table). The above protocol ensures a uniform walk-through among all the AR experiences. However, our protocol does not cover some cases, such as locked menus or pay-to-continue schemes (e.g., gaming). We analyse all the apps according to the following steps: (1) the app and interface were globally scanned. (2) anything that was odd-looking in the app was noticed. (3) all findings were listed in a table, and (4) were grouped into different classes. (5) the current taxonomy of MDs and the grouped classes were compared. (6) it was decided whether each individual grouped class could be determined as an MD or as a misleading advertisement according to [17, 40].

**Main findings.** We categorise the observed MDs used in the collected MAR apps. The first category, *surroundings*, includes all findings related to apps requiring the user to scan any part of the environment to use an AR feature. The survey found cases where the software required the user to scan a room, a part of a room, a wall, or any part of an environment. The second category, *filters*, includes all findings related to any visual change that differs from the actual camera perspective on the user's device. During the survey, cases occurred where layers were blurred, the lighting or colours looked different, the skin was smoothed, and beauty filters were applied. These differences were detected by comparing pictures of the MAR in the app with pictures of the same visual. Only the beauty filters can be detected when using the app. The other filters/filter-like findings could only be detected by comparing the pictures. The third category, *scaling*, includes all findings related to objects or features changing their shape. In the survey, cases were found in which the size of furniture and shoes were made either bigger or smaller. In addition to these findings, more differences were detected by comparing pictures of the AR in the app with pictures of the same visual. While doing this, another case was found that made the lips of the user look bigger. The fourth category, *requests*, includes all findings that require the user to do any action for using the AR feature. Inquiries are required using the microphone, enabling notifications, or looking into the camera to take action. One finding was that the app required the user to smile into the camera to continue.

## 4 USER STUDY

The above results reveal the current MD techniques in MAR apps and further exploration is necessary, as the recorded approaches do not take all the advantages of AR environments, such as immersion, realism, and mobility. Thus, another study evaluates the effects of MDs. We implemented two fully working MAR apps, where the

---

[1]The list of apps collected: *Furniture/interior*, American furniture warehouse, Macy's furniture, Home depot, Wayfair, Target, Magnolia market, Amazon, Anthropologie, Coolblue, Houzz, Ikea place; *clothing/accessories*, Nike fit, Ray Ban-Virtual Try On, Crocs, Ideofit, Lacoste LCST, Converse, Charlie Temple, Tissot, Warby Parker, GAP, POIZON, Tiffany & co wedding rings, Zara AR app concept by Supersuper agency, Zara Shop the look, Topshop, Uniqlo augmented reality app, Timberland augmented reality mirrors, Poshmark-try on glasses, Gucci-try on shoes, Valentino -try on shoes; *beauty*, Sally, Hansen, Sephora, L'Oreal virtual makeup app, Wanna nails, Macy's beauty (in-store not in an app), Ulta beauty, Too Faced, Walmart-try on makeup, NYX-try on makeup.

**Table 2: Study scenario set-up.** *MD* stands for the scenario with manipulative design in the right or left item (flowerpots/cats) and NoMD, the scenario without MDs.

| App | Conditions | Side | Sound description | Object and sounds combination |
|---|---|---|---|---|
| Flowerpot | manipulative design (MD) | Right | Screeching bird sounds(1) and Calming sea waves(2) | Left flowerpot (1), Right flowerpot (2) |
| Flowerpot | MD | Left | Screeching bird sounds(1) and Calming sea waves(2) | Left flowerpot (2), Right flowerpot (1) |
| Flowerpot | no-manipulative design (NoMD) | - | No sound(3) | Left flowerpot (3), Right flower pot (3) |
| Cat | MD | Right | Hissing of a cat(4) and Meowing of a cat(5) | Left cat (4), Right cat (5) |
| Cat | MD | Left | Hissing of a cat(4) and Meowing of a cat(5) | Left cat (5), Right cat (4) |
| Cat | NoMD | - | Same (pos) sound(6) | Left cat (6), Right cat (6) |

participants can select between two items they would like to buy (Figure 2). The 1st and 2nd apps display two flowerpots and cats of different colours, respectively.

**Experiment design.** A factorial study design compares the effects of MDs on the participants' decision-making (Table 2): two scenarios (between-participants: two MAR apps) × two sides (between-participants: unpleasant sound on the right or left object) × two conditions (within-participants: manipulative design (MD), no manipulative design (NoMD)).

**Experimental conditions.** The rationale for the MD design is grounded on the characteristics of AR settings and the use of auditory feedback to impact the participants' decision-making [23, 43, 49]. Table 2 describes all the apps' conditions. Two audio sounds are included for each flowerpot in the first app: (i) screeching bird sounds and (ii) calming (sea) wave sounds. The screeching bird sounds (MD) can influence the participants to select the other flowerpot (which includes calming (sea) wave sounds). The MD for any of the flowerpots was assigned to each participant. Both flowerpots are silent in conditions with no MDs (NoMD). In the second app, a similar technique follows: two different cats own the following audio feedback: (i) the hissing of a cat and (ii) the meowing of a cat. Both applications activate the sound when the participants are near the object (a cat or a flowerpot). *A Latin Square* alleviates the carryover effects by randomizing the order of the conditions (MD, NoMD) [36]. This also applies to allocating the MD sound for the right and left sides. As our main focus was the audio effects on the participants' responses, we randomised the visual designs of flowerpots and cats accordingly.

**Participants and apparatus.** Forty participants were recruited (9 Females, 30 males, and 1 non-closure), aged 18 to 44 years old. Amongst the participants, 87.5% of them were 18-24 years old, while 10% and 2.5% were 25-34 and 35-44 years old, respectively. 72.5%, 17.5%, and 10% of them earned bachelor's, master's, and doctoral degrees, respectively. The participants rated their previous experience with MAR apps: 15% excellent, 47.5% good, 27.5% average, and 10% poor. 24 participants (60%) mentioned that they are familiar with MAR apps through games, social media, Google Maps, or measurement apps. The participants evaluated their confidence in MAR applications as 3.48 (SD: 0.84) on a scale of 1 to 5 (maximum). An iPhone 11 Pro was employed for the two MAR applications. We use *Adobe Aero* [2] to create our applications, which produce a clean and engaging interface. We instructed the participants to wear headphones to receive audio from the applications. The experiment was done in a quiet office inside the university campus.

**Task and procedures.** The experiment leverages participants' choice between two augmented objects to examine the effect of

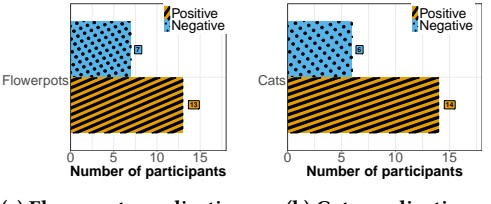

(a) Flowerpots application.  (b) Cats application.

**Figure 3: Participants' responses according to the application used – flowerpot (left) & cat (right), and the item with positive (calming sea waves & meowing) or negative (screeching & hissing) sound selected.**

using MD designs. We used a successive sampling method to invite participants to our laboratory. Before the experiment, we obtained their agreement by having them complete a consent form that outlined our data-gathering and processing practices. Upon obtaining the participants' agreement, we instructed them to move into the designated position inside our laboratory, as indicated by markings on the floor. Next, we instruct the participants to launch the augmented reality applications on their iPhone 11 Pro devices and put on their headphones. The participants were instructed to engage in the augmented reality (AR) environment, which allowed them unrestricted movement. An instructor provided instructions on engaging with the flowerpots and cats on the screen and encouraged the users to approach the things before selecting. Subsequently, we instructed the participants to indicate their preferences for the flowerpot and cat by physically approaching the desired item and interacting with the corresponding images on the smartphone's screen. According to the scenario configuration, the flowerpot & cat (selected by the participants) will or will not include audio feedback when they are in proximity. After interacting with each scenario, we asked the participants whether they could spot any malicious design in the MAR app, where the participants can answer 'noticed', 'not noticed' and 'not sure'. If the participants answered 'noticed' and 'not sure', we asked them to describe the manipulative design (MD) they found in the scenario. Finally, we asked the participants some demographic questions (i.e., age, gender, education), their relevant experience with AR apps, and their trust in AR apps, following a 5-point Likert scale [45] (1-very low to 5- very high).

### 4.1 Key Result: Impact of MDs in MAR

**Choice of item.** Figure 3 depicts the participants' responses in the scenarios with MDs and NoMDs. After satisfying the normality and homogeneity, we perform a Chi-square test to examine the

**Table 3: Participants' responses regarding their perception of manipulative designs (MDs) in MAR applications. The positive sound nudges the object selection with more pleasant audio feedback (e.g., wave sounds in the flowerpot application).**

| Application | Order | Choice: pos./neg. sound | MD design | | | Participants' MD descriptions |
|---|---|---|---|---|---|---|
| | | | Noticed | Not noticed | Not sure | |
| Flowerpot | MD - NoMD | pos. 80%(n=8) | 63%(n=5) | 25%(n=2) | 13%(n=1) | All the participants(n=8) designated the sound as MD. |
| | | neg. 20%(n=2) | 50%(n=1) | 50%(n=1) | 0% | 50%(n=1) designated the sound as MD. |
| | NoMD - MD | pos. 50%(n=5) | 100%(n=5) | 0% | 0% | All the participants(n=5) designated the sound as MD. |
| | | neg. 50%(n=5) | 60%(n=3) | 20%(n=1) | 20%(n=1) | 40%(n=2) designated the sound, while 20%(n=1) designated the sound in conjunction with the starting point as MD. |
| Cat | MD - NoMD | pos. 70%(n=7) | 43% (n=3) | 28% (n=2) | 28% (n=2) | 57%(n=4) designated the sound as MD |
| | | neg. 30%(n=3) | 33% (n=1) | 0% (n=0) | 67% (n=2) | 67%(n=2) designated the sound as MD |
| | NoMD - MD | pos. 70%(n=7) | 57% (n=4) | 43% (n=3) | 0% | 67%(n=4) designated the sound as MD |
| | | neg. 30%(n=3) | 67% (n=2) | 33% (n=1) | 0% | 67%(n=2) designated the sound as MD |

effect of the MDs in the participants' selection. We found that a significant effect of the manipulative designs on the participants' responses ($\chi^2(1) = 6.0842$, $p < .05$) exists. When the participants were faced with the application with manipulative design, 67.5% (n=27) of them selected the item (flowerpot & cat) with the more pleasant sound (i.e., calming or meowing sound). Compared with the NoMD scenarios, only 40% (n=16) of the participants selected the same item with a positive sound.

**Response time per condition.** A paired samples t-test, satisfying the normality assumptions, shows no statistical significance in the effect of MDs on the response time to select a flowerpot ($t(19) = 0.55$, $p = 0.588$) or cat ($t(19) = 1.55$, $p = 0.138$). The use of MDs in our applications does not increase participants' response time

**Location of the MD.** A Chi-square test reveals the absence of MD location effects on the participants' decision ($\chi^2(1) = 2.12$, $p = 0.1446$), indicating all MD shares similar effects, regardless of item locations (left or right item).

**Comparison of applications 1 & 2.** The different visualisations of flowerpots and cats do not have an impact on the participants' selections. A Chi-square test shows that there is no effect of displayed item type (flowerpots or cats) on the participants' decisions ($\chi^2(1) = 0.113$, $p = 0.7357$). This finding shows the possibility of generalising similar MD designs with other scenarios (e.g., different objects, menu interfaces).

**Order of conditions.** We perform a Chi-square test that shows that there is no correlation between the order we show the MD condition (1st time or second time) and the participants' responses ($\chi^2(1) = 1.025$, $p = 0.311$). The MD scenario has the same impact on the participants' decisions, despite being the 1st/2nd time they have to use the application.

## 4.2 Key Result: User Perceptions to MDs

**Discoverability of MDs.** Table 3 summarises the results of the study. The majority of participants notice the audio feedback as MDs. Our results show that the majority of the participants who selected the pleasant audio feedback in the MD scenarios and noticed a malicious design detected the audio feedback as the MD technique. 59% (n=16) of the participants that selected the pleasant sound objects (67.5%, n=27) detected the audio feedback MD. Overall, 72.5% (n=29) of the participants detected the MD as audio feedback provided in MD scenarios. In the NoMD scenarios, 27.5% (n=11) of the participants noticed MDs such as objects in the background (e.g.,

trees, table), the distance between the objects (flowerpots & cats), and the scale of the objects. Most participants refers to the sea wave sound as a "likeable" or "enjoyable" sound, while the screeching bird sound was often described as "scary", "loud" or "uncomfortable". Some participants mentioned not noticing the difference in the flowerpot colour, because of their distraction towards the sounds, i.e., audio becomes more dominant than visuals [54].

**Influence of MAR experience on detection of the MD.** The MAR experience of the participants does not affect the effectiveness of the MDs ($\chi^2(3) = 1.084$, $p = 0.780$). The participants with 'excellent' and 'good' experiences with MAR apps (62.5%, n=25) have similar percentages of detecting the MDs (72.5%) than the ones that rated their experience as 'average' or 'poor' (37.5%, n=15).

## 5 COUNTERMEASURES AND VALIDATIONS

Section 4 showed that we could use audio to manipulate users in a metaverse shopping environment. Non-visual cues can be considered vulnerable parts of a metaverse shopping experience design. We attempt to search for countermeasures to prevent manipulation in the non-visual cues of a metaverse experience design [17]. Instead of sole self-assertions, we collect user feedback and the acceptance levels of the proposed countermeasures. Several virtual scenarios were proposed to users to help them determine their preferred metaverse shopping experiences and understand their willingness to implement the proposed countermeasures. If the users are willing to implement the countermeasures, we anticipate that they can be implemented successfully. We surveyed the user's perception of the countermeasures, which aims to clarify the users' expectations regarding metaverse commerce. Questions primarily cover manipulative designs hidden in audio, feelings for metaverse shopping experiences, agreement and disagreement on the features that may appear in future AR/VR shopping, and the users' opinions towards threats and countermeasures. We propose several topics and methods inspired by previous works: 1. *Traditional two-factor authentication* [3]. 2. *Voiceprint verification* [8]. 3. *Visual output constrain for AR/VR application* [30][31]. 4. *Firewall system* [19]. 5. *Malicious web link detection* [13][27]. Inspired by the findings in Section 4, audio becomes a potential threat as a non-visual manipulative design. Therefore, the user survey contains questions addressing audio-driven e-commerce scenarios. Accordingly, we build scenarios that depict the possibilities of manipulation in both visual and non-visual cues in a metaverse shopping experience.

**Table 4: Questions for Threats Scenarios (Odd #, TS) & Countermeasures (Even #, CT) in 4 Domains (D), with mean values & standard deviations (SD). The Pair Sum (PS) is an aggregated mean of a paired TS and CT, i.e., user perceptions of a TS & its CT.**

| # | D | Question | M | SD | PS |
|---|---|---|---|---|---|
| 1 | 1 | How would you feel about shopping in a store with AR product displays on your mobile device? | 2.82 | 1.19 | |
| 2 | 1 | Before displaying the aforementioned AR product displays on your mobile devices, a simple privacy statement should be explained. | 2.95 | 1.22 | 5.77 |
| 3 | 3 | How would you feel about a virtual shopping market including virtual avatars that give information? | 2.78 | 1.19 | |
| 4 | 3 | Before talking to a virtual avatar, the device you are using should provide a pop-up with a yes/ no statement linked to a warning about your voice data being recorded. | 2.76 | 1.42 | 5.54 |
| 5 | 2 | How would you feel about the usage of sounds in a Metaverse shopping experience? | 2.65 | 1.06 | |
| 6 | 2 | The usage of sounds in a Metaverse shopping experience should be announced beforehand. | 3.34 | 1.10 | 5.99 |
| 7 | 1 | How would you feel about a display of virtual products that changes at different time intervals based on your interest? | 2.73 | 1.23 | |
| 8 | 1 | Cameras that track your eyes and body movement to decide your interests/ preferences for products should visually show this tracking on a monitor for more awareness. | 3.24 | 1.30 | 5.97 |
| 9 | 2 | How would you feel about being able to interact with virtual items that are displayed for try-out? | 2.55 | 1.30 | |
| 10 | 2 | When clicking on items, a malicious VR item warning should be given. | 2.91 | 1.29 | 5.46 |
| 11 | 3 | Would you like to have the 3D salesman for AR/VR shopping experience? | 2.59 | 1.17 | |
| 12 | 3 | Before the salesman starts talking to you, you should personally permit to enable speech or sounds coming from the salesman. | 3.00 | 1.26 | 5.59 |
| 13 | 4 | Would you want to purchase a product in a virtual store using technologies like Apple/Samsung in a virtual store? | 2.62 | 1.19 | |
| 14 | 4 | Before purchasing in a virtual store, you should be asked to use the two-factor verification to think about making your purchase once more. | 3.04 | 1.26 | 5.66 |
| 15 | 3 | How would you feel about a virtual shopping assistant helping you by advising you about products in an AR/VR environment? | 2.73 | 1.21 | |
| 16 | 3 | Would you like to have a firewall system for detecting the source of an AR/VR agent (which may block your interaction)? | 3.00 | 1.29 | 5.73 |
| 17 | 4 | How would you feel about finding secure ways of paying in the Metaverse? | 2.57 | 1.33 | |
| 18 | 4 | Would you agree with the use of voiceprint verification in AR/VR for payment and personal data protection? | 3.20 | 1.23 | 5.77 |

**Domains of the metaverse shopping experience.** The survey questions (Table 4) can be categorised into four specific domains (and the relevant questions) in the metaverse experience: (1) immersive ways for product display (Questions (Q) 1, 2, 7, and 8), (2) virtual environment (Q5, 6, 9, and 10), (3) virtual avatar (Q3, 4, 11, 12, 15, and 16), and (4) payment methods (Q13, 14,17, and 18). These domains (1 – 4, the 2nd column in Table 4) can represent possibilities for the user to be manipulated. We cluster the survey questions into these domains to exemplify a structured result in all the possible scenarios. We had an internal discussion among the authors of the paper. To avoid a lengthy questionnaire, we designed and selected 2 to 3 pairs of threat scenarios and countermeasures for each domain, based on the likelihood of the threat concerns advised by the paper's senior authors. We acknowledge that the questionnaire of 18 questions can only cover a limited set of threats and one of many countermeasures.

**Participants and survey process.** Our survey begins by asking the participants to disclose their experience related to AR and VR, age, and familiarity with audio. Next, the survey continues by asking the participants about their feelings, attitudes, and awareness of using immersive technologies with audio cues. To understand the participants' perceptions of virtual scenarios and countermeasures, we resume the survey by presenting the user with 18 questions (Table 4) regarding metaverse shopping experiences and countermeasures for possible manipulations in these experiences. Following the Likert 5-point scale, the rating of all survey questions is divided into five levels of experience (Very positive; Positive; Neutral; Negative; Very negative) with the metaverse, in terms of *the experience of online shopping*, which probes the most acceptable countermeasures that receive a positive rating in the survey.

We recruited 100 participants for the survey, with the following demographics: the participants' AR/VR experiences ($\bar{M}$= 1.34 years, $\sigma$= 0.47), age groups ($\bar{M}$= 24.3, $\sigma$= 1.30), and familiarity with audio ($\bar{M}$= 2.95 (out of 5), $\sigma$= 1.25). We acknowledge that the participant background could affect the result validity, i.e., ecological issues. Also, the participants reported more than neutral (i.e., slightly positive) averaged ratings towards feeling ($\bar{M}$= 2.74, $\sigma$= 1.15), attitude ($\bar{M}$= 2.65, $\sigma$= 1.21), and awareness ($\bar{M}$=2.98, $\sigma$=1.28). Furthermore, we highlight the pairs of traits in which correlation exists.

**Age and Experience** The correlation coefficients between attitude and experience exist, age: $r_{att,exp} = -0.28$, $r_{att,age} = -0.16$, i.e., the attitude has a weak negative relationship with experience and age. Similarly, the correlation coefficients between awareness and experience are: age: $r_{aw,exp} = 0.19$, $r_{aw,age} = 0.19$. Thus, age variation and user experience do not significantly affect the attitude towards countermeasures and awareness of malicious design.

**Familiarity and Feeling** The correlation coefficients between attitude and familiarity, attitude and feeling, awareness and familiarity, and awareness and feeling are $r_{att,fam} = -0.64$, $r_{att,feel} = 0.81$, $r_{aw,fam} = 0.80$, $r_{aw,feel} = -0.49$. These coefficients reveal strong negative relationships exist among the paired traits of attitude and familiarity, awareness and feeling, and very strong positive relationships between the pairs of attitude and feeling, awareness and familiarity. The above statistical analysis implies the casual relationships: (1) With a more positive feeling towards auditory AR/VR, the users are less likely to notice the manipulation and demonstrate a more positive attitude toward the countermeasures. (2) While users are more familiar with auditory AR/VR, the users are more sensitive to the manipulations but do not show much interest in the countermeasures.

**Summary of Survey Results** We elaborate on the results per type of reaction (positive/ neutral/ negative) of the threats and countermeasures, as shown in Table 4. Among all the questions, most respondents made the "very positive" to "neutral" responses to our established scenarios. The 4th and 5th columns of Table 4 list the averaged values and the standard deviations of the participant responses per question. We mention the standard deviation reaction per domain to indicate the spread of the values of the survey results. The results can be compared among either the odd-numbered or even-numbered questions to understand if participants are less likely to accept a particular countermeasure because of the participants' attitudes towards the initial threat scenario. For example, the first pair of questions (Q1 and Q2) belongs to Domain 1, namely immersive product display; "How would you feel about shopping in a store with AR product displays on your mobile device?". The participants commented slightly positively on the threat scenario (Q1) and countermeasure (Q2), resulting in mean values of 2.82 and 2.95, respectively. Accordingly, we aggregate the two mean values

of the pair, denoted as Pair Sum (PS), indicating the appropriateness of constructing the most acceptable countermeasure to address the corresponding threat scenario.

Furthermore, we report the mean values of threat scenarios and countermeasures by categories, as follows. *Immersive product display (1)* shows that the mean values are slightly higher than 2.5, indicating that participants are open to accepting the scenarios and countermeasures. Participants are more used to these scenarios and are likely to understand them more easily. *Virtual environments (2)* contain Questions 5 & 6, which have the lowest standard deviation of each question pair, which could be related to the simplicity of the questions. These questions consider only audio and the announcement of sounds, which should be familiar to almost all survey participants, regardless of their experience with the metaverse. *Virtual avatars (3)* have the highest numbers of averaged values among each domain, ranging between 2.59 and 3.00. Participants are interested in adding these features to their experiences, but they also feel an even greater need for good protection in these scenarios. *Payment methods (4)* comprise the situations and experiences with the most favourable replies. This is expected to become a common technique for immersive applications. Nonetheless, this area has the most unfavourable reactions to the countermeasures. Consequently, it may be argued that such settings are readily manipulable, especially through non-visual clues. Despite contemplating this risk within the metaverse environment, participants still opt to utilise it less securely due to the possibility that their interactions would be restricted excessively.

**Top-ranked countermeasures & threats** According to the survey results, we ranked the feedback on the countermeasures with a heuristic approach. The pair with the highest pair sum (the 6th column) is the sum of mean values reflecting the question pair of a threat scenario and a countermeasure (Table 4). The countermeasure with the highest acceptance rate can serve as a user-acceptable solution to protect the user's rights or vice versa. Next, we choose the paired threat scenario and countermeasures with the highest pair sum in each domain. For instance, the paired mean value of Q1 (2.82) and Q2 (2.95) is 5.77 but slightly lower than that of Q7 and Q8, resulting in 5.97. Thus, Q7 and Q8 are selected in the domain of immersive ways for product display. The first countermeasure that appears per domain in the ranking is anticipated to have the highest chance for acceptance in its domain. Each best countermeasure in its domain is as follows. For *immersive product displays (1)*, the best countermeasure is notifying the user by physically showing the eye movement tracked on a monitor. With *virtual environments (2)*, the best countermeasure is announcing incoming sounds in the experience. Regarding *virtual avatars (3)*, firewall warning about the source of a virtual agent. Finally, voice-print verification is the best countermeasure for *payment methods (4)*.

To make the vision of each best countermeasure per domain more comprehensible, we illustrate several threat scenarios and complementary countermeasures to address the threats. Each of these scenarios corresponds to one of the four domains to give an understanding of the ways countermeasures could work in real-life situations. We strive to create more understanding amongst the user concerning the countermeasures so that in future scenarios, users can understand its needs. Thus, when the user encounters

one of the countermeasures in the real world, they are less likely to resist the countermeasure and gain protection.

*(1) Immersive ways of product display (Q7 & Q8).* This scenario presents a virtual avatar, a sales representative who displays a virtual presentation of various products to a user. The virtual salesperson wears a company badge with a hidden camera (Figure 4a). With this camera, the company can track the users' eye movement to detect the products that are most preferred by users. As for the countermeasure, Figure 4b alerts the user of eye movement tracking by displaying the detected patterns. The user acts on the information and goes shopping for clothes elsewhere.

*(2) Virtual environments (Q5 & Q6).* Figure 4c highlights how a milking factor misleads a user who attempts to find a restaurant for lunch. Two restaurants, denoted as A and B, are close to the user. The user prefers restaurant B over A but would settle with whichever restaurant they encounter due to hunger. Additionally, the user can view and hear the navigation cues through the see-through AR smartglasses. Meanwhile, a verisimilitudinous AR barrier and a subsequent prompt of '*turn right*' guide the user to restaurant A. In other words, the user would miss out on restaurant B by following the prompt, which is complemented by the bird's-eye view of the user's position (the top-left corner, Figure 4b).

Accordingly, we present the countermeasure (Figure 4b) in which the user's device can distinguish the virtual objects from the real world and annotate the virtual-physical blended objects. An algorithmic approach can search whether the immersive overlays cause a substantial impact on the user's behaviour; for instance, a roadblock or picket line might cause the user to re-route in real life. As such, a visual warning appears in front of the user asking the user to allow the incoming sound and notifying the users that the barrier is a non-real entity with malicious purposes, i.e., leading to a detour in this navigation scenario. The proposed countermeasure can guarantee a threat-free experience and save the users from unwanted interruption that influences their choices.

*(3) Virtual avatars (Q15 & Q16).* Figure 4e illustrates a potential threat of someone adopting a sales-like avatar to trick users into oversharing their personal information, especially when the avatar looks very realistic, even though it has a fraudulent identity in reality. The virtual salesperson claims that he is an accredited collector through his AR avatar, but turns out to be an impostor. The avatar with a professional outlook provides illusions, and further lures the classical music fan to pay a premium for a fake composer's manuscript. The countermeasure refers to a firewall-based detection of fraudulent input (Figure 4f). A system screens the inputs and alerts the users once malicious acts are detected.

*(4) Payment methods (Q17 & Q18).* Figure 4g illustrates a scenario where a user is shopping for a tote bag in AR. The bag comes with audio that makes the user think about the beach and summer even though it is winter. The AR experience plays into the users' sentiment by creating the illusion of a nice summer day. In this way, the user might be tricked into buying an object it does not need. Figure 4h mentions a voiceprint verification approach that offers the user additional time to think about their purchase decisions, this way the user has an extra moment out of the AR experience to rethink their choice for buying the product.

(a) A virtual salesman selling clothing to a user through AR.

(b) Warning the user about potential AR eye-tracking detection.

(c) An object in AR obstructs the user from going to the preferred direction.

(d) The incoming sound needs to be enabled because of the malicious object.

(e) A sales avatar tricking a user into sharing personal information.

(f) The user receives a notification that the source cannot be detected.

(g) A shopping bag in AR that includes non-visual cues.

(h) The two-factor verification allows the user to rethink their purchase.

Figure 4: Pictorial description of the malicious design and potential countermeasures (enlarged visuals available in [4]).

## 6 CONCLUSION & DISCUSSION

It is crucial to highlight the potential hazards in the Metaverse, provide guidelines for the future design of e-commerce, and suggest ways to safeguard users from potential dangers in the MAR environment. We conducted a series of user studies to better understand the possible user threats and propose countermeasures. Audio feedback, especially with DPs, can influence the user's behaviours. Our examples shed light on the ethical design with non-visual cues in the virtual-physical blended world. Meanwhile, we explored the emerging threats and understand the manipulation scenarios that users can encounter, although our work is limited to a small number of examples. In the remainder of the paper, we provide a concise discussion of possible design insights for metaverse e-commerce.

**Virtual agents.** Our experiment (Figure 2) shows agents have an important role in enhancing the shopping experience. Augmentations, such as visuals and audio, can help customers explore the products in more detail, and provide more straightforward product purchasing. For example, the increased adoption of voice assistants (e.g., in smart homes) allows users to purchase items, such as food and house products, by using their voice. In the metaverse, such approaches can also be implemented in the virtual world using virtual agents using AI. These agents (voice assistants for AR, virtual agents for VR) can still manipulate the shoppers' purchase decisions using, e.g., different tones and pitches. Moreover, in the case of VR, the visual appearance of these agents can also affect the social dynamics [53] and, therefore, users' purchase behaviours.

**Multimodal MetaShopping.** The metaverse will leap forward in terms of customer experience and interaction modalities. Compared with more traditional systems (e.g., desktops, smartphones), the metaverse can include visual, audio, and haptic feedback to the customer experience. For example, haptic interactions can communicate and affect the users' emotions in the metaverse [5, 6, 34, 52]. In this work, we have seen audio effects on users' perceptions of MDs and their influence. The multimodal interaction possibilities that the metaverse can bring to customers can enhance the experience and challenge the perceptions of more sophisticated MDs (e.g., using audio, visual and haptic interactions simultaneously).

**Dynamic spaces.** The e-commerce of the metaverse will bring more ubiquitous interactions with the use of technologies, such as AR and VR. Grounded in the scenarios in [18], we can see how the metaverse can bring new interaction paradigms, such as proxemics. In these scenarios, the physical locations will have different areas where customers can interact with virtual assets such as virtual agents (e.g., voice assistants) or AR interfaces. Virtual spaces can also bring more dynamic shops, where the content surrounding the avatars can change according to their personal preferences (e.g., mood, hobbies, style). These dynamic spaces also open new threats to users regarding MDs, where sellers can adapt their spaces according to the mood of the shoppers and influence their decisions. In the physical world, sellers usually analyse the customers' reactions and behaviour to achieve the most efficient way to sell a product. In the metaverse, AI-based agents could predict the shoppers' behaviour and emotions with high accuracy. The metaverse will bring a new shopping experience but not without higher risks of users being manipulated. Therefore, researchers, regulatory organisations, and developers should be fully involved to avoid equivalent practices. There remains optimism that these parties will be taking an active role and protecting the future customers of the metaverse, perhaps before the destructive practice becomes the major norm. We consider CCPA a good starting point for governing such manipulation.

**Design guideline.** We identify the emerging types of malicious design that may severely impact users in the immersive scenarios of e-commerce. Figure 5 connects the threat issues and countermeasures and suggests several design guidelines for developers and interaction designers, including *Audio, 3D Interfaces, and 3D Avatar*, to secure the consumer from privacy leakage and financial loss in their daily activities in virtual-physical blended environments.

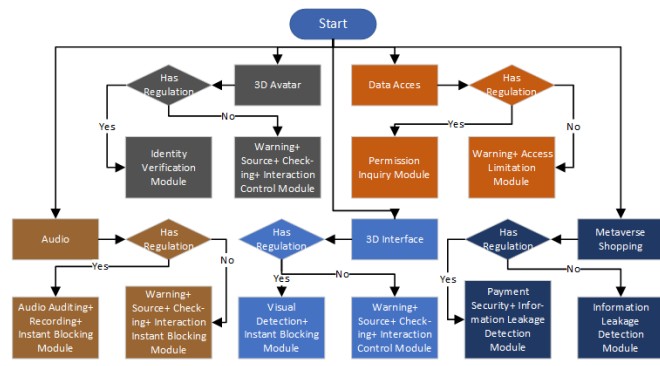

Figure 5: The design guideline of countermeasures for malicious designs in the metaverse.

# ACKNOWLEDGMENT

This research is supported by the Hong Kong Polytechnic University's Start-up Fund for New Recruits (Project ID: P0046056), and a grant from the Guangzhou Municipal Nansha District Science and Technology Bureau under Contract No.2022ZD012.

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
