# OpenReview forum: "Towards Trustworthy MetaShopping: Studying Manipulative Audiovisual Designs in Virtual-Physical Commercial Platforms"
_acmmm.org/ACMMM/2024/Conference — MM2024 Oral_

### Official Review · Reviewer_6GND · 2024-05-16

**Rating:** 4
**Confidence:** 1

**Summary:**

In this paper, the author conducted a series of user studies to investigate manipulative designs in VR commercial applications. The experimental results demonstrated that audio and visual elements can more effectively convey product details to customers.

**Strengths:**

- This work is interesting and it contributes to commercial design, which may make sense in smart homes, metaverse, and so on.
- The author provides a comprehensive overview of the development of e-commerce and highlights the widespread use of VR/AR at the current stage.

**Limitations:**

- The number of scenarios (Table 2) presented in the experiments of this study is limited, which may not effectively demonstrate comparative results.

**Suitability:**

3

---

### Official Review · Reviewer_dWz6 · 2024-05-21

**Rating:** 5
**Confidence:** 4

**Summary:**

The paper explores the impact of Manipulative Designs (MDs) in XR e-commerce applications.

**Strengths:**

Well written paper describing a study which was well designed, conducted, and evaluated in an area of interest to the community.

**Limitations:**

As a paper submitted to the research track of a prestigious and highly selective conference such as ACM MM, one expects there to be an underpinning (and novel) research question. This is missing in the paper. However, the study design is sound and can easily accommodate at least one research question - this needs to be explicitly stated in the paper, and the analysis and discussion structured accordingly (one might argue that "What is the impact of Audiovisual MDs in MAR shopping?" could be a candidate research question in which case the paper does not need restructuring. Anyway, I leave it to the author(s).

In respect of the 40 participants taking part in the study, the sampling strategy needs justifying as well as the recruitment strategy employed. It is also unclear if ethical approval had been obtained from the relevant bodies in order to conduct research.

In respect of multisensory VR/AR/Xr shopping, perhaps the author(s) might be interested in the following papers:

Comşa, I. S., Saleme, E. B., Covaci, A., Assres, G. M., Trestian, R., Santos, C. A., & Ghinea, G. (2019). Do I smell coffee? The tale of a 360 mulsemedia experience. IEEE MultiMedia, 27(1), 27-36.
Mishra, A., Shukla, A., Rana, N. P., & Dwivedi, Y. K. (2021). From “touch” to a “multisensory” experience: The impact of technology interface and product type on consumer responses. Psychology & marketing, 38(3), 385-396.
Rahman, M. S., Bag, S., Hossain, M. A., Fattah, F. A. M. A., Gani, M. O., & Rana, N. P. (2023). The new wave of AI-powered luxury brands online shopping experience: The role of digital multisensory cues and customers’ engagement. Journal of Retailing and Consumer Services, 72, 103273.

One is also surprised that the following seminal work is uncited:

Cialdini, R. B. (2001). The science of persuasion. Scientific American, 284(2), 76-81.

As a minor point, the listing of reference [16] needs correcting, as the author's name seems to appear multiple times "[16] Maya Georgieva, Maya Georgieva Maya Georgieva is an EdTech...."

**Suitability:**

3

---

### Official Review · Reviewer_tRDm · 2024-06-09

**Rating:** 4
**Confidence:** 2

**Summary:**

Immersive technologies and the Metaverse have the potential to become a game changer for e-commerce. However, appropriate Manipulative Designs (MDs) or dark patterns, among other relevant aspects, need to be incorporated to fully exploit such a potential.
The paper contributes to exploring MDs that take advantage of the immersive metaverse (e.g., realism and multi-modal cues), e.g., visuals and audio, in augmented e-commerce scenarios. A series of related user studies have been conducted to shed some light in this context. It includes a user study (n=40 participants) is conducted for two AR apps, demonstrating that audio feedback as MDs can significantly influence users’ purchasing decisions, and design guidelines are provided based on the lessons learned. In addition, design guidelines are provided, based on the lessons learned

**Strengths:**

- The paper addresses a very relevant and timely topic, with commercial impact.
- Three relevant user studies are conducted, all relevant findings are provided.

**Limitations:**

- The paper provides a comprehensive number of references, and the related work section categorizes different relevant aspects for the topic addressed in the paper. However, lists of references are provided for specific aspects / concepts, but the main insights from such references could be further elaborated in Section 2.
- Immersive technologies and mediums are diverse and provide new and sophisticated multi-modal means for interaction and exploration, compared to traditional digital platforms. The paper focuses on AR settings, which are very relevant in the context of e-commerce, but it could also include a brief reflection on intrinsic implications and/or opportunities brought by immersive tech / mediums when it comes to MDs applied to e-commerce applications.
- The paper could have paid more attention to privacy / ethics aspects.
- Section 3: Title is “Manipulation of Existing Apps” but the main goal is to identify existing apps, test them and derive lessons learned from such study. Then, I would suggest re-phrasing the title, and to elaborate further on pros / cons / innovative features from existing apps beyond just identifying MD categories. Usability and immersion aspect could also had been considered.
- The message of the abstract and introduction seem to be more oriented on understanding the impact of MD when using AR apps for e-commerce, while the message of the conclusions section focuses more on understanding possible user threats and propose countermeasures. This should be better linked or homogenized.
- While immersive mediums allow for integrating rich multi-modal MDs, as discussed, the paper mostly focused on rather basic audio-driven MDs. This is acceptable, as the paper needs to be focused and audio is relevant, but further discussion on the relevance of the focus of the paper should be provided and stronger contextualization should be provided..
- While the paper highlights in the abstract and introduction that it provides design guidelines, these are mostly centered on countermeasures for malicious designs, but not on effectively adding multimodal MDs.
Section 4:
- The selection of these 2 apps should be justified
- “We use Adobe Aero [2] to create our applications” -> it should be clarified whether the two selected AR apps were already existing or were developed by authors
- The apps and considered MDs are quite simple; and further it is unclear their relevance to the e-commerce sector. This should be discussed and justified. Authors should also discuss – ideally with evidences – the impact of audio feedback stimuli on users’ attitudes and behaviour in e-commerce scenarios.
- “This paper proposes the design space of manipulative techniques in e-commerce applications for the metaverse.” -> It is true that the paper reports on three relevant user studies, discusses lessons learned, and proposes design guidelines mostly centered on countermeasures for malicious designs, but I would not state that a design space for MD in MAR is provided. Authors should clarify and discuss that, if accepted.
Open Questions:
- The apps were used on smartphones. What would be the implications of using AR/XR headsets? At least, it should be discussed.
Typos / Writing Issues:
- Section 3 -> three major categories, so I guess number (3) needs to be added for beauty.
- Three researchers examine -> examined
- Please, check the writing all over the paper.
- All acronyms need to be defined the first time they are introduced (e.g. XR in Section 2), and then only the acronyms should be used (e.g., AR).

**Suitability:**

3

---

### Meta-Review · Area_Chair_ydUx · 2024-07-04

**Recommendation:** Accept (Oral)
**Confidence:** 4

**Metareview:**

The paper received two Boderline Accept and one Weak Accept Recommendations, which have been kept after having analyzed the rebuttal by authors. Thus, the paper can be accepted for presentation at ACM MM 2024, but authors are encouraged to address the reviewers’ comments as much as possible in the final version of the paper